# *Xylella fastidiosa* subsp. *pauca*, *Neofusicoccum* spp. and the Decline of Olive Trees in Salento (Apulia, Italy): Comparison of Symptoms, Possible Interactions, Certainties and Doubts

**DOI:** 10.3390/plants12203593

**Published:** 2023-10-17

**Authors:** Marco Scortichini, Giuliano Manetti, Angela Brunetti, Valentina Lumia, Lorenzo Sciarroni, Massimo Pilotti

**Affiliations:** 1Research Centre for Olive, Fruit Trees and Citrus Crops (CREA-OFA), Council for Agricultural Research and Economics (CREA), 00134 Rome, Italy; marco.scortichini@crea.gov.it; 2Research Centre for Plant Protection and Certification (CREA-DC), Council for Agricultural Research and Economics (CREA), 00156 Rome, Italy; giuliano.manetti@crea.gov.it (G.M.); angela.brunetti@crea.gov.it (A.B.); valentina.lumia@crea.gov.it (V.L.); lorenzo.sciarroni@crea.gov.it (L.S.)

**Keywords:** olive tree, *Xylella fastidiosa* subsp. *pauca*, *Neofusicoccum mediterraneum*, *Neofusicoccum stellenboschiana*, Botryosphaeriaceae, olive quick decline syndrome—OQDS, branch and twig dieback—BTD, dieback, leaf tip necrosis, cigarette-like leaf rolling

## Abstract

*Xylella fastidiosa* subsp. *pauca* (XFP), *Neofusicoccum mediterraneum*, *N. stellenboschiana* and other fungi have been found in olive groves of Salento (Apulia, Italy) that show symptoms of severe decline. XFP is well known to be the cause of olive quick decline syndrome (OQDS). It has also been assessed that *Neofusicoccum* spp. causes a distinct disease syndrome, namely, branch and twig dieback (BTD). All these phytopathogens incite severe symptoms that can compromise the viability of large canopy sectors or the whole tree. However, their specific symptoms are not easily distinguished, especially during the final stages of the disease when branches are definitively desiccated. By contrast, they can be differentiated during the initial phases of the infection when some facets of the diseases are typical, especially wood discoloration, incited solely by fungi. Here, we describe the typical symptomatological features of OQDS and BTD that can be observed in the field and that have been confirmed by Koch postulate experiments. Similar symptoms, caused by some abiotic adverse conditions and even by additional biotic factors, are also described. Thus, this review aims at: (i) raising the awareness that declining olive trees in Salento do not have to be linked a priori to XFP; (ii) defining the guidelines for a correct symptomatic diagnosis to orient proper laboratory analyses, which is crucial for the application of effective control measures. The possibility that bacterium and fungi could act as a polyspecies and in conjunction with predisposing abiotic stresses is also widely discussed.

## 1. Introduction

### 1.1. The Decline Syndromes in Apulia: Unravelling an Intricate Matter

*Xylella fastidiosa* (XF) is a xylem-limited Gram-negative Gammaproteobacterium that infects more than 600 plant species and spreads in plant communities by way of xylem-feeding hemipteran insect vectors [1,2]. Though XF mostly has an endophitic behavior, it is well known to cause devastating diseases in some plant species, such as, for example, grapevine, peach and citrus species, which suffer, respectively, from the so-called Pierce’s Disease (PD), Phony Peach Disease (PPD) and Citrus Variegated Chlorosis (CVC) [3,4,5].

In October 2013, XF was reported in Salento (Apulia, Italy) [6] on about 10.000 ha of olive groves (*Olea europea* L.) that showed severe twig and branch diebacks [7,8]. Soon after, the disease was named “olive quick decline syndrome” (OQDS) [8]. All XF isolates in Salento harboring the same sequence-type, ST-53, were classified as XF subspecies *pauca* (XFP) and were genetically highly similar to XFP isolates from central America, thus suggesting a single introduction event from that world area [9,10,11]. In addition to the bacterium, the initial surveys ascertained the occurrence of some fungi and insects in OQDS-affected trees, namely, the xilophagous insect *Zeuzera pyrina* excavating galleries in the wood at the larval stage, as well as the fungi *Phaeoacremonium* spp., *Phaeomoniella* spp., *Pleurostoma* sp. and *Neofusicoccum* sp. [8,12], all isolated from discolored wood. These first reports did not present details on the number of the examined specimen trees or the frequency of the fungal species. Simultaneously, investigations were conducted into a decline of olive trees in the northern part of Apulia—the Canosa di Puglia, Cerignola and Foggia areas—thus geographically distinct from Salento, the Apulian area that was found to be infected with XFP [13]. Surveys of a representative number of olive trees and reports of fungal frequency data highlighted the presence of several fungal taxa, but *Pleurostoma richardsiae*, *Neofusicoccum parvum* and diverse species within *Phaeoacremonium* were revealed as the most frequent, and given their virulence, the etiological role was especially attributed to *P. richardsiae* and *N. parvum* [13,14]. Interestingly *P. richardsiae* was able to kill all inoculated twigs in a timeframe of 25 days after the inoculation. An activity aimed at detecting XF in the examined trees was not reported, but the area of this survey is still XFP-free nowadays.

Due to the quarantine status of the bacterium for the European and Mediterranean Plant Protection Organization [15], most of the studies performed after the OQDS outbreak concerned XFP. Moreover, the role(s) played by such fungi in OQDS was judged as not relevant [16], despite their occurrence in olive trees also infected by the bacterium [12,17,18], their capacity to reproduce disease symptoms when artificially inoculated [13], their finding in areas close to the front of the OQDS expansion [19] and their production of metabolites that are phytotoxic to olive [18]. A confirmation that factors other than XFP are involved in the severe decline of olive trees in Apulia—i.e., possibly fungi—clearly arises from the data from monitoring surveys obtained by Regione Apulia during 2017–2018 in the “containment” and “buffer” areas located northward of the initial areas of the XFP outbreak, showing the absence of the bacterium in 3.300 olive trees that still showed typical symptoms of OQDS [20].

It has been well established for a long time that several diseases of tree species are not attributable to single agents causing them consistently; rather, they are the result of sequential, combined and cumulative effects of multiple factors. Synclair and Hudler [21] and Manion [22] discuss this matter, describing emblematic case studies and theorizing four decline concepts under which the different case studies could be categorized. The third concept is the most popular and is representative of how a decline is established on a tree, in that it includes a sequence of interchangeable factors—predisposing, inciting and contributing—each of which enables the pathogenic action of the subsequent one and ends with tree death. Climate, site and biotic factors fall into the three categories. Importantly, none of the factors involved in a decline is significantly pathogenic on its own. A number of climate and site factors have been associated with OQDS in Salento, and their possible role within the decline syndrome has been discussed [23,24]. A further development of the decline concepts and complex diseases is represented by a growing body of studies that highlight how different pathogenic microorganisms can contribute to causing a crop disease through synergistic interactions that are triggered during co-infections [25,26,27]. Moreover, in some circumstances, weakly pathogenic or nonpathogenic microbes are also mutualistically involved in causing the disease. This occurs in the presence of predisposing factors, with the microbes acting as a polyspecies, as, for example, in the acute oak decline [27].

Concerning woody species, an example of an interaction between a primary pathogenic species and microbial endophytes is represented by the European ash (*Fraxinus excelsior*) disease, in which the fungus *Hymenoscyphus fraxineus* is the main causal agent [28]. However, it has been observed that the diversity and composition structure of the microbial communities of European ash leaf are strongly linked with the severity of the disease. In addition, the host genotype also influences leaf fungal community composition [29]. Among olive diseases, there is evidence of interactions between different bacterial species causing specific symptoms. In the olive knot, indeed, there are synergistic interactions between bacterial species that are non-pathogenic for olive trees, such as *Erwinia toletana*, *E. oleae* and *Pantoea agglomerans* and the main pathogen causing the knot, *Pseudomonas savastanoi* pv. *savastanoi.* In fact the knots derived from co-infections are larger than those caused by *P. savastanoi* pv. *savastanoi* alone [30,31].

### 1.2. The Case of Botryosphaeriaceae and Branch and Twig Dieback in Salento

Recently, in Salento, fungal species of the Botryosphaeriaceae family, namely, *Neofusicoccum mediterraneum* and *N. stellenboschiana,* have been isolated from olive trees affected by symptoms that are apparently attributable to OQDS. Twig desiccation and progressive dieback were among the most common symptoms [32,33]. Pathogenicity trials demonstrated the capacity of these fungal species to cause lethal bark canker, wood discoloration and twig wilting. Thus, their involvement in the observed symptomatology was clearly demonstrated—to such an extent that we decided to name the disease branch and twig dieback (BTD). It is worth remembering that various Botryosphaeriaceae species, such as *N. mediterraneum*, *Diplodia mutila*, *D. seriata* and *Neoscytalidium dimidiatum*, have been reported as primary agents of severe decline, characterized by twig and branch dieback, canker and wood discoloration, in olive trees in diverse world countries—California (USA), Spain, Croatia and Turkey [34,35,36,37,38]. Very recently, pathogenic isolates belonging to different species within *Neofusicoccum*, *Botryosphaeria* and *Diplodia* were isolated from declining olive groves in Uruguay [39]. Specifically, the decline described in Spain and California, in which *N. mediterraneum* and *D. mutila* were the most aggressive Botryosphaeriaceae agents, was called twig and branch dieback [36,37]. Interestingly, the presence of XF in olive trees had been previously reported in California in olive trees affected by leaf scorch and dieback [40,41]. Soon after the study by Úrbez-Torres et al. [36] that established the fungal etiology of this decline, the work by Krugner et al. [42] definitively assessed that *X. fastidiosa* subsp. *multiplex* (XFM) infected olive trees in California (USA). However, an association with branch and twig dieback was ascertained in 16.6% of 198 sampled trees. Moreover, inoculation tests showed that XFM was not detected after inoculation in most cases, even after one year. In very few cases, detection was positive but only transiently. Out of a high number of inoculated plants, two test plants had stably positive results, i.e., for the entire observation period (one year). In all cases, no symptoms were expressed by the inoculated plants. These results prompted the authors of the study to conclude that the olive tree is not a preferred host for XFM, whose infection, when it occurs, may be strongly self-limiting, such that chronic infection may be uncommon. Obviously, this indirectly confirmed the fungal etiology ascertained by Úrbez-Torres et al. [36] for twig and branch dieback of olive trees in California.

At a first glance, indeed, the symptoms that can be observed on an olive tree infected with Botryosphaeriaceae in Salento strongly recall those incited by XFP, namely, OQDS. However, there are some differences that can be counted, especially during the initial phase of the disease, which can be observed with the naked eye [32,33]. Ongoing field surveys, coupled with subsequent laboratory analysis on tree samples, are allowing us to ascertain that in Salento, there are cases of olive trees affected by OQDS and BTD-like symptomsand infected by: (i) XFP solely (i.e., the fungi were not isolated from the specimens); (ii) Botryosphaeriaceae (i.e., XFP was not detected in the specimens); and (iii) XFP and Botryosphaeriaceae (i.e., the bacterium was detected, and the fungi were isolated), which seems to be the most frequent case [43].

These findings depict a more complicated epidemiological scenario concerning the decline of olive trees in Salento. Indeed, one might wonder whether: (i) BTD and OQDS are two distinct diseases, frequently overlapping with additive effects; (ii) the respective agents contribute to disease expression according to a *sensu strictu* decline concept, namely, the disease state arises from the concerted action of predisposing, inciting and contributing factors [21]. It is worth noting that the latter hypothesis does not necessarily exclude the fact that bacterium and fungi are also able to act separately as pathogens, and Koch postulate experiments have in fact demonstrated this [32,33,44]. Importantly, this speculative scenario of causality would not underplay the possible roles of environmental abiotic stressors and the host plant reactivity to external insults/stimuli [23,24]. It should be noted, indeed, that a single olive tree apparently infected solely with XFP takes quite a long time to express symptoms and completely wilt, i.e., anywhere from two to five years [45]. Moreover, there is an awareness that some XFP-infected olive trees tolerate the infection without expressing any symptoms for a very long time, suggesting that additional factors are necessary to undermine this resilient condition and trigger the disease expression [46]. We still do not know the exact impact of co-infections of XFP and *Botryosphaeriaceae*—and possibly other fungi—in causing decline, but the high virulence expressed by *N. mediterraneum* and *N. stellenboschiana* in the pathogenicity tests carried out with olive plants would point to a relevant role for these fungi [32,33].

The contemporary occurrence of XFP and Botryosphaeriaceae also has implications in the management of XFP-caused OQDS. An effective control strategy has been set up to reduce the XFP cell density within the canopy of olive trees that allows for their maintenance and production also in the infected areas of Salento [47,48]. In the case of co-occurrence of fungi, this strategy should be coupled with additional measures specifically targeted at preventing or restricting fungal invasion.

The aim of this review is to provide descriptions and images that can help to distinguish symptomatic features linked to XFP and those directly caused by Botryosphaeriaceae. First of all, this has a scientific value in raising the awareness that declining olive trees in Salento have not to be linked *ex ante* to XFP; secondly, a correct symptomatic diagnosis, followed by proper laboratory analyses to confirm the identity of the causal agent(s), will be the basis for a rational approach to applying effective control measures. As suggested above, we also retain crucial and necessary to discuss on the likelihood and reasonability of different scenarios of interactions between XFP and Botryosphaeriaceae, given that OQDS and BTD are geographically overlapping, and XFP-plus-Botryosphaeriaceae mixed infections frequently occur.

## 2. The Overall Appearance and Progress of Olive Declines in Salento

Mature olive trees progressively desiccate starting from twigs basipetally, thus according to a dieback mode. This symptomatic pattern can first affect distinct sectors of the canopy and then extend successively to the whole tree. Thus, death of the tree is generally the final outcome of the symptomatology, although sometimes, resprouting at the collar can occur (Figure 1).

In severely declining trees, growers attempted to halt the dieback by severe pruning. Nevertheless, this measure did not work, and the disease had a lethal outcome (Figure 2) [23].

In some cases, super-high-density (SHD) plantings with olive trees cv FS-17 (synonym Favolosa), which are considered resistant/tolerant to XFP [49], replaced the destroyed groves (Figure 2). Observations over time will definitively verify the real reaction of this cultivar to the decline.

Inexplicably, we found some olive groves or even distinct trees nearby to those severely affected that were completely asymptomatic and in optimal vegetative condition, though they were infected with XFP, or their infectious state had not been possible to ascertain due to the ban imposed by the growers (Figure 3).

## 3. *Xylella fastidiosa* subsp. *pauca* (XFP) in Olive Tree: Symptoms and Parallels with Pierce’s Disease

Here, we describe distinctive traits of XFP-incited symptoms and discuss the lifestyle of the bacterium—commensalism vs. parasitism—in connection with the olive tree and in parallel to some aspects of grapevine PD caused by XFF.

### 3.1. Symptoms Caused by Xylella fastidiosa subsp. pauca (XFP) (Olive Quick Decline Syndrome—OQDS)

The natural symptoms that we describe here refer to olive trees ascertained for the occurrence of XFP in the analyzed diseased specimens taken from the trees. The initial leaf symptom is leaf tip desiccation (i.e., leaf scorch), which progresses toward the petiole end of the leaf blade, slightly quicker in proximity of the midrib (Figure 4 and Figure 5). This symptom is the most distinctive apart from the full-blown stages of the disease (i.e., twig and branch diebacks and plant death). However, it is worth noting that leaf tip desiccation is not a pathognomonic feature of the XFP infection in olive trees. Indeed, water stress, salty winds, the absorption of heavy metals and boron deficiency can induce similar symptoms [50]. We also observed leaf tip desiccation on declining XFP-uninfected olive trees in the Latium region [43].

In artificially infected small-sized olive trees cv Cellina di Nardò (40–60 cm in height at the moment of inoculation), Saponari et al. [44,51] demonstrated the following: XFP took as long as 9 and 12 months, respectively, to perform a consistent systemic colonization (reaching 18 cm above the inoculation point) and a complete colonization of the whole plant including the roots. Most of the inoculated plants were symptomatic at 12–14 months after the inoculation (mpi). In two-year-old olive trees of the same cultivar (two meters tall at the moment of the inoculation), XFP was detected at 10–15 cm above the inoculation point at 7 mpi, but it was also found in uninoculated shoots. Symptoms appeared in all replicates 8–10 months after the inoculation, and desiccation and dieback were generalized at 24 mpi. Overall, this demonstrates that XFP takes a long time to colonize systemically the plants of the susceptible cultivar, and even more slowly it incites the wilting symptoms; namely, after colonization is completed, an additional incubation time is necessary for symptoms to be expressed. Specifically, in the inoculated plants, the following symptoms were described: “leaves became first chlorotic, then withered, turned brown and desiccated. Symptoms consistently started from the apical portion of the inoculated shoots and progressed toward their base” [44]. The photographs included in the report of 2016 add further details in that they show that both leaf tip desiccation and, more frequently, a homogeneous withering of the leaf blade occur. Moreover, it is also clear that the leaf blade rolls downward—resembling a cigarette—during desiccation or even earlier, during the phase of leaf chlorosis. Overall, the importance of leaf tip desiccation as a symptom linked to XFP infection in olive tree is indirectly confirmed by the pathogenicity trials performed with XFP in *Polygala myrtifolia,* which evidently showed leaf tip desiccation rapidly scorching the whole leaf blade. Pathogenicity trials performed by Saponari et al. [44,51] also highlight that the strongest symptomatic response was strictly associated with the highest bacterial population inside the plants and the capacity to systemically colonize the plants, especially the roots. In this respect, cv Cellina di Nardò proved the most susceptible compared with cvs Coratina, Frantoio and Leccino, in which the systemic movement and multiplication rate of the bacterium as well as symptom expression were delayed and reduced. This clearly suggests that the host species harbors a genetic variability that is able to condition the phenotype of susceptibility.

The above results refer to trials conducted in a greenhouse at a controlled temperature regime. Interestingly, in the same investigation, a parallel trial was conducted in natural uncontrolled conditions under a net tunnel. All cultivars used in this trial (the same as cited above) showed a poor colonization in terms of systemic movement and multiplication rate, and they did not show any disease symptoms throughout the whole course of the experiment (24 months). This strongly suggests that not only the host genotype but also environmental factors, possibly temperature fluctuations, can deeply influence the outcome of the infection events.

In Argentina and Brazil (South America), XFP was reported on olive trees in 2015 and 2016, respectively [52,53]. The associated symptoms consisted of leaf chlorosis, different degrees of leaf scorching starting from the tip of the leaf blade, twig wilting and severe desiccation of the branches. In successive reports by the Brazilian research group [54,55], symptomatology was confirmed, and PCR detection applied on different symptom types revealed that leaf tip desiccation symptoms were consistently associated with XFP (82% of samples were indeed XFP-positive). On the contrary, twigs with ongoing cigarette-like wilting were negative. The presence of wood discoloration in small branches of the affected trees was also photographically documented. Germplasm collections were XFP-positive, though asymptomatic. Inexplicably, the inoculation trials conducted over a period of 30 months did not yield any symptoms, though the bacterium was detected in some of the inoculated olive trees. The most common sequence type of XFP found in Brazil in olive tree and used for pathogenicity tests is ST16. This sequence type has some genetic differences compared with ST53, which is spread in Apulia; thus, any claims of parallelism with the Apulia outbreak should be cautious. Nevertheless, until the Koch postulate is fulfilled, the exact etiological role of XFP ST16 in the observed olive symptomatology remains to be established. A survey conducted to evaluate XFP distribution and variability in different locations in the southeast area of Brazil enabled the detection of four XFP STs (ST16, 84, 85, 86) in 43.7% of the sampled olive trees showing OQDS-like symptoms [56]. The lack of a full association between symptoms and bacterium could be interpreted as a real lack of a causative relation between XFP and the observed decline in that country, or it may be that additional stresses can cause the OQDS-like disease in Brazil in addition to XFP.

Investigations into wood discoloration and the associated fungal microbiota have not been documented/conducted in Argentina or Brazil so far [54,55,56]. On the contrary, in Uruguay, an olive decline was investigated throughout the country that was associated with virulent Botryosphaeriaceae, as above mentioned; unfortunately, information on the presence of XF was not reported. Interestingly, Uruguay borders the southeastern part of Brazil, in which the olive decline has been surveyed for XFP presence, as described above [39].

All things considered, etiological investigations into the olive decline in South American countries would greatly benefit from the parallel detection of both XF and fungi.

On the basis of world reports that date XF-infected olive trees back to 2004 in California and 2013, 2015 and 2016 in Italy, Argentina and Brazil, respectively, it seems that the system of XF-olive tree was naturally established quite recently. All the results reported above suggest that the relationship of XF with olive tree ranges between non-preferred, self-limiting asymptomatic infections, chronic asymptomatic infections and infections whose symptomatic outcome seems to depend on a variety of factors that have still to be elucidated. In this respect, the host plant genotype, environmental conditions, bacterial subspecies and sequence type would play important roles in conditioning the interaction between the bacterium and the olive tree.

### 3.2. Brief Outline of Pathogenicity of Xylella fastidiosa subsp. fastidiosa in Pierce’s Disease of Grapevine: Can It Be a Lesson?

*Xylella fastidiosa* subsp. *fastidiosa* (XFF) is the cause of Pierce’s Disease (PD), one of the most devastating diseases of cultivated grapevine, to date restricted to the Americas [3]. Given the economic importance of the host plant, a large body of scientific studies have contributed to increasing the knowledge on this pathosystem. This model can thus be used to orient the investigation into other XF systems. Here, we briefly outline the processes underlying XFF progression within the host and the onset of PD symptoms, for the purpose of drawing some parallels with the olive-XFP pathosystem.

It is well known that, within the host plant, XF manifest in: (i) an aggregative biofilm-forming immobile phase in which clusters of cells are kept together, encapsulated within a matrix of macromolecules that robustly adhere to xylem vessels, and proceeding to (ii) a planktonic, explorative, motile, single-celled phase in which XF is more prone to spreading in the vascular system and to multiplying [57]. The first condition seems to guarantee a protection against the host immune system and would thus increase the ability of the bacterium to establish chronic and stable infections; not less importantly, the bacterium forms the “sticky” biofilm phase for efficient vector acquisition and transmission. The second condition is instead linked to virulence, as planktonic cells express proteins involved in cell wall and pit membrane degradation, which enables vessel-to-vessel movement; it has also been shown that a mutant form of XF—hypermotile, hyperproliferating and impaired in biofilm formation—is hypervirulent [58]. This latter finding is in agreement with findings in olive tree, according to which extensive spread and multiplication rate is linked to severe symptom expression in artificially inoculated olive trees [44]. Work by Sun et al. [59] showed that the biofilm-forming and the planktonic bioforms coexist in both susceptible and resistant varieties of grapevine. In olive tree cultivars susceptible to OQDS—Ogliarola and Cellina di Nardò—both bacterial phases were also observed, i.e., rare and solitary bacterial cells in the xylem vessels of the branches and large sessile cell aggregates in the xylem of foliar petioles [60,61]. However, a regulation of the two bioforms occurs in order to tune the relationship with the host plant: commensalism vs. parasitism, asymptomatic vs. symptomatic. Solitary XF cells in a low-density environment produce high amounts of spheroid outer membrane vesicles (OMVs) containing membrane materials that are released outside and prevent the attachment of the bacterial cells to each other and to plant surfaces, thus contributing to the maintenance of the single-celled exploratory phase. On the contrary, as local cell density increases, another factor becomes crucial upon progressive accumulation: the diffusible signal factor (DSF), which induces a high number of adhesins promoting cell stickiness and represses factors involved in cell motility, i.e., plant cell wall-degrading enzymes (PCWDs), type IV pili and OMVs. Hence, overall, DSF promotes the biofilm sessile phase [57]. Indeed, work by Scala et al. [62] clearly suggests that olive tree has also the potential to regulate the biofilm-forming immobile phase vs. the planktonic explorative phase by differentially expressing a profile of lipid signals—fatty acids and derived oxylipins—which impacts on phase promotion/inhibition. A fairly established sight, though still to be fully demonstrated, is based on the fact that XF generally has a commensal relationship with most of its host plant species, in which the exploratory phase would be tightly self-limited (just enough to disseminate sticky bacterial plaques in diverse points of the vascular system in order to maximize the stochastic encounter with the insect vector?). This implies that a robust homeostasis guarantees an appropriate mutual adaptation between the host plant and the microorganism. It has been suggested that an XF–plant species interaction devoid of the proper adaptation, as in the case of XFF–grapevine, gives rise to a disease and represents a commensal relationship “gone badly” [63]. This view would imply that in PD, the exploratory phase is enhanced compared to any other purely commensalistic relationship. However, also in this condition, some regulation still occurs to allow time to ensure bacterial survival through multiplication and insect transmission before the current host plant is killed [57,59,63]. However, comparative studies on the behavior of XF in asymptomatic and symptomatic host plants are needed in order to confirm this view.

In grapevines, inoculation of XFF requires from two to four months to achieve evident symptom development and a detectable XFF systemic colonization [59,63,64,65,66]. Although this is considered a long time for a Koch postulate experiment to attain a response, the case of Koch postulate experiments with XFP–olive tree far exceeds this time, taking over a year in the Apulian case, and yielding no symptomatic response in both the Brazilian and the Californian cases. If we apply the theory described above regarding the perturbation of XF lifestyle from a basic commensalism vs. parasitism, it seems that XF infections in olive tree worldwide are strongly self-limiting, ranging from a commensalism that is poorly infectious to a very long incubation-based parasitism in which the extremely slow movement and the delayed symptom expression are a clear remnant of the commensal self-limiting behavior of XF, much more pronounced than in PD-affected grapevine [42,44,53,54,55]. This statement only apparently conflicts with the high in-field damage of olive trees in Salento, especially if we consider our introductory remarks on the etiological complexity of this damage. Specifically, we retain an interesting hypothesis that Botryosphaeriaceae might somehow exacerbate the dysregulation of XFP commensalism versus parasitism as well as directly contributing to dieback, as previously demonstrated [32,33]. Interestingly, analysis of the microbiota residing in the cane endosphere of XFF-infected grapevines, either symptomatic or asymptomatic—the latter named disease escape vines—revealed that *Pseudomonas fluorescens* and *Achromobacter xylosoxidans* showed significant negative correlations with XFF titer, suggesting that these bacteria might be a potential driver of the disease escape phenotype [67]. Importantly, Botryosphaeriales, mostly *Diplodia*, were also detected, but correlations with the disease phenotype were not reported in this paper. In light of this finding, the above cited hypothesis on the role of Botryosphaeriaceae in connection with XFP acquires further strength, though it predicts an opposite effect as that suggested in grapevine infected with XFF, *P. fluorescens* and *A. xylosoxidans.*

## 4. Symptoms Caused by *Neofusicoccum mediterraneum* and *N. stellenboschiana* (Branch and Twig Dieback, BTD)

Trees affected by BTD show twig wilting and dieback, resulting in the death of branches. The observation of natural wilting symptoms at different stages enabled us to establish the following symptom progression: initially, red-bronze patches are scattered on the leaf blade or start from the leaf edge or the midrib. Occasionally, they are surrounded by a chlorotic halo. Subsequently, the lesions spread, necrotize and coalesce, thus affecting the entire leaf blade, which rolls downward, resembling a cigarette. In some cases, rolling occurs before necrosis appears and together with a generalized chlorosis of foliage (Figure 6). In Figure 7, an overview of olive trees with severe BTD is shown.

At the base of the wilting twigs or branches, the bark shows sunken, necrotized areas that are reddish to brown in color, with scattered blackish spots, clearly suggesting a canker with a basipetal progression (Figure 8). Correspondingly, the xylem shows a discoloration pattern that can be either diffuse/scattered or wedge-shaped in the most evident cases (Figure 9). This suggests that a dysfunction in water conduction may act distally, leading to water leaking from leaf blades, which is not compensated for by an adequate transport in the xylem vessels.

Observations of the affected branches indicate that: (i) the primary necrosis generally starts from the bark, extending longitudinally and, importantly, tangentially, thus tending to girdle the whole circumference of a twig/branch. Simultaneously, from the bark, necrosis spreads into the xylem radially, likewise via medullary rays. Hence, the severity of xylem discoloration is strictly linked to the rate at which necrosis girdles the bark. Depending on whether less than, equal to or more than 50% of the bark circumference has been girdled by necrosis, the so-called wedge-shaped cankers are geometrically shaped as a “sector of a circle” that is less than, equal to or greater than a “semicircle”, respectively (Figure 10).

Sampling twigs and branches of BTD-affected trees revealed that wood discoloration is not always obvious, though bark necrosis is generally more evident. In fact, wood discoloration can be restricted longitudinally, instead appearing widespread transversally, just enough to create a dysfunction in the distal portion of the twig/branch. Moreover, discoloration may not be found right below the wilted parts but even in the proximal older branches bearing them. In other cases, the connection between wilted parts and discolored wood is direct and evident.

Pathogenicity trials performed with *N. mediterraneum* and *N.stellenboschiana* fulfilled the Koch postulate, namely, the natural symptoms observed in the olive groves were reproduced [32,33]. Both fungal species were able to wilt one-year-old twigs, but *N. mediterraneum* was the most virulent in terms of repeatability and the short time taken for the onset of wilting (15–20 days for a complete desiccation after the inoculation, but leaf rolling started much earlier). Long necrotic streaks were caused at the base of two- to three-year-old stems affecting both the bark and wood. The necrotic patterns were clearly wedge-shaped cankers: those caused by *N. mediterraneum* extended repeatably in a cross-section to more than a semicircle, with a girdling index—the ratio between the tangential spread of bark necrosis and the stem circumference—between 0.62 and 1.00, whereas those caused by *N. stellenboschiana* were repeatably more restricted, being roughly less than or equal to a semicircle, with a girdling index between 0.38 and 0.60 (Figure 10 and Figure 11).

The wood discoloration patterns caused by these fungal species in the inoculation trials perfectly matched those observed in the olive trees naturally affected by BTD.

Interestingly, both *N. mediterraneum* and *N. stellenboschiana* expressed their virulence regardless of the time of the artificial inoculation (spring/summer and autumn/winter), suggesting that they are fully pathogenic to the olive tree for most of the year. Overall, the results of the pathogenicity trials demonstrate the causal relationship between these fungal species and natural symptoms and also suggest that *N. mediterraneum* and *N. stellenboschiana* probably work together as a polyspecies in inciting/contributing to the BTD olive decline [32,33]. It is not to be excluded that further investigations in Salento will highlight the presence of additional Botryosphaeriaceae species that would thus enlarge this pathogenic polyspecies.

## 5. Similar Symptoms Caused by Additional Biotic Agents

Upon closer look, biotic agents other than XFP and Botryospaeriaceae can cause disease symptoms similar to those described for these pathogens, but with features enabling a distinction.

### 5.1. Parasitic Brusca

Parasitic brusca is characterized by necrosis of the tip and, differently from leaf tip desiccation caused by XFP, also other areas of the leaf blade. The agent is the fungus *Marthamyces panizzei* (ex *Stictis panizzei*) (teleomorph). After pathogen entry through the stomata in autumn, the mycelium colonizes the mesophyll, forming apothecia in winter and spring on the adaxial leaf surface. Apothecia are easily visible to the naked eye; they appear like tiny black dots scattered in the necrotized tissue, and this also helps to support a fungal etiology. Pycnidia are formed in autumn in the abaxial leaf surface, at the first appearance of foliar symptoms, and they release conidia, which perpetuate infection events [68,69,70].

### 5.2. Leaf Necrosis and Fruit Rot Caused by Neofusicoccum luteum

*Neofusicoccum luteum*, another species of Botryosphaeriaceae, was reported in different regions of Australia as the causal agent of leaf tip desiccation as well as fruit rot of olive tree in 2003–2008. The most damage was recorded in 2008, following an unusually wet summer. Similar to parasitic brusca, fungal structures—pycnidia, in this case—appear embedded in the necrotic leaf tissue and are visible to the naked eye as black dots, thus helping orient the diagnosis, most likely fungi [71].

### 5.3. Canker and Leaf Scorch of Olive Tree Caused by Neoscytalidium dimidiatum

The fungus *Neoscytalidium dimidiatum* belongs to Botryosphaeriaceae and is considered the main cause of a severe and widespread olive decline in the southeast Anatolia region of Turkey [38]. Trees are affected by dieback of the scaffold—twigs, branches and even the stem. Bark canker and wood discoloration manifest. Death can occur in diseased trees, especially in young trees up to 15 years old. Twig and branch dieback is associated with foliage wilting, with cigarette-like rolled leaves in scattered sectors of the canopy, similar to what is observed in OQDS- and BTD-affected trees. Artificial inoculation showed that this was the effect of stem/branch/twig infection by the fungus. An additional distinctive feature of the symptomatology is the apical leaf scorching, namely, leaf tip desiccation, turning dark yellow to brown, “with a clear separation between the necrotic and healthy tissues”, spreading toward the petiole and giving rise to extensive defoliation. Spray inoculation of non-wounded leaves demonstrated that the fungus can directly infect the leaves and cause the apical leaf scorch, with the development of blackish pycnidia on the abaxial surface of the leaves. Interestingly, pathogenicity trials also showed that this leaf symptom is cultivar-specific, as just one cultivar (the Turkish Gemlik), among the seven cultivars tested, developed this symptom [38]. The authors of the study discussed the similarity of leaf tip desiccation observed on *N. dimidiatum*-infected trees to that of XFP infections and underlined that the development of fungal structures on the leaf blade was a discriminating factor. Nevertheless, today, considering that XFP and potentially pathogenic fungi can co-infect the same trees, it is necessary to verify the presence of XFP with molecular diagnostic methods whenever and wherever a decline phenomenon is investigated in olive tree.

### 5.4. The Effects of Toxic Compouds Produced by Fungi Colonizers of Olive Tree Wood

Bruno et al. [18] studied the effects on olive tree of culture filtrates obtained from liquid cultures of *Celerioriella prunicola*, different species of *Phaeoacremonium* and *N. parvum* isolated from olive trees cv Ogliarola Salentina, infected with XFP and showing symptoms of OQDS. Detached twigs were soaked with their base in the culture filtrates and observed for two weeks. Filtrates of all fungal isolates incited clear toxic effects such as leaf chlorosis, withering and browning. In particular, the cigarette-like leaf rolling symptom was incited by culture filtrates of all fungal isolates, thus proving once more to be a non-specific disease sign.

### 5.5. Viral Infections Causing the “Leaf Yellowing Complex”

To date, sixteen defined viral species and an additional two viruses, still unassigned, have been identified in olive tree, most of which were found to be associated with symptomless trees or for which the link with the symptomatology still remains to be investigated [72,73]. However, with regard to the so-called “leaf yellowing complex”, its viral etiology seems to be convincingly ascertained [72]. This symptom complex corresponds to three different diseases, which are “olive vein yellowing” (OVY), “olive leaf yellowing” (OLY) and “olive yellow mottling and decline” (OYMD). The typical symptoms of the complex are poor fruit set, bright yellowing of the foliage, leaf mottling, extensive defoliation and dieback [72,74]. Specifically, the first disease, OVY, has been found to be associated with olive vein yellowing-associated virus (OVYaV) [75], and the infected olive trees showed vein yellowing and yield reduction resembling the disease reported in central Italy by Ribaldi with the name of “infectious yellows” [76] (Figure 12). The second disease, OLY, has been associated with olive leaf yellowing-associated virus (OLYaV) [77]. This virus is widely spread in olive and in some cases can cause a bright yellow discoloration of the leaves [73,77]. The third disease, OYMD, is caused by olive yellow mottling- and decline-associated virus (OYMDaV) and is characterized by leaf mottling, yellowing and necrosis starting from the leaf tip, leading to severe leaf fall and dieback [72,77]. Considering the aforementioned features—chlorosis, foliar necrosis and dieback—only a rough similarity with OQDS and BTD can be found.

## 6. Similar Symptoms Caused by Abiotic Agents

Adverse conditions incited by abiotic factors that can persist over middle- or long-term periods can cause leaf tip desiccation symptoms similar to, but not to be confused with, those incited by XFP [50].

### 6.1. Symptoms Caused by Hot and Salty Winds

Hot winds, sometimes accompanied by salty particles from the sea, are quite frequent in Salento, especially during the end of spring and summer, and it has been noted that they can cause damage to the olive trees, namely, a severe leaf tip desiccation and even defoliation. The necrotized leaf tissue appears completely dry, greyish and shriveled contrasted with the remaining green part of the leaf, and this aspect does not match with those caused by XFP (Figure 13a). This phenomenon can be reminiscent of the non-parasitic brusca that was described (together with parasitic brusca) at the end of 18th and beginning of the 20th centuries and that occurred in Salento. It was reasoned that non-parasitic brusca was caused by physical factors affecting the water content of leaves, such as: quick alternation of hot and cold periods, hot dry winds and salty winds. In general, brusca caused severe damage mainly to the eastern areas of the Lecce province, causing relevant leaf tip desiccation and eventually complete leaf scorch, defoliation and twig dieback. However, investigations were not conclusive, and conclusions drawn by the diverse pathologists in those historical times were also conflicting (see Frisullo et al. [78] for an exhaustive revision of the matter). At the beginning of the XFP outbreak, the leaf tip desiccation made scientists hypothesize the involvement of brusca in the OQDS. However, soon after, this hypothesis was discarded [78].

### 6.2. Symptoms Caused by Boron Deficiency

Boron deficiency is not rare to olive trees. Lack of this microelement can cause leaf chlorosis, necrosis of the leaf tip (Figure 13b), increased leaf thickness, leaf distortion, twig dieback accompanied by the occurrence of secondary shoots that lead to a “rosetta” formation and twig bark protuberance accompanied by phloem necrosis [50]. The combination of these symptoms allows for distinguishing boron deficiency from either XFP or *Neofusicoccum* spp. However, in many circumstances, the severity of boron deficiency can be moderate, and leaf chlorosis is the sole symptom that refers to it and enables one to distinguish it from potassium deficiency.

### 6.3. Symptoms Caused by Potassium Deficiency

In the olive groves of the Mediterranean areas, potassium deficiency is frequently found. It negatively influences the water use efficiency and growth of the olive tree [79]. The main symptoms are a smaller size of the leaves, which also show tip necrosis, usually without any sign of chlorosis (Figure 13c). In addition, the leaf tip curves downward, whereas the twigs show shorter internodes.

### 6.4. Symptoms Caused by Winter Frost

Winter frost, although not very frequent, can occur in the Mediterranean areas of olive cultivation. Severe leaf scorching and defoliation occur at −7 °C, and the tree, depending on the cultivar’s susceptibility, dies around −12 °C [80]. Split of the bark and the xylem in branches and trunk frequently occurs upon a frost event [81]. Recently, in Salento, winter frost events have caused damage to the olive trees in addition to that caused by OQDS [23].

Leaf tip desiccation typical of XFP infections must not be confused with marginal leaf scorching, which appears after winter frost (Figure 14). However, in advanced stages, marginal leaf scorching evolves in a total collapse and desiccation of leaf blades. This can progress from the distal part of the blade towards the petioles.

### 6.5. Symptoms Caused by Waterlogging (Experimentally Reproduced)

Waterlogging, due to sudden and repeated pouring rain, is becoming increasingly frequent in recent times, being strongly connected with climatic changes. The intensity and frequency of rain can cause devastating landslides and flooding of towns and the countryside [82,83]. Specifically, olive groves in Salento have been repeatedly subjected to waterlogging after extreme precipitation events [23]. The time of flooding cannot be evaluated solely on the basis of the time of permanent stagnant water appearing above the ground. Indeed, a deep flood in the ground can have a more prolonged duration. The permanence of water saturation just for some days establishes a condition of hypoxia in the soil, which in turn produces hypoxia in the roots. In general, in plants, hypoxia causes a switch from an aerobic metabolism, able to regularly fulfill ATP synthesis through the oxidative phosphorylation, to an anaerobic metabolism, which resorts to fermentation to satisfy energy requests. Although the induction of the fermentative metabolism is a requirement for basal tolerance, prolonged waterlogging causes the accumulation of toxic compounds such as lactic acid, alcohols, aldehydes and other anaerobic metabolites, ultimately leading to plant death [84].

We mimicked the effect of waterlogging on young olive tree plantlets by packaging the substrate-root bread with plastic bags and filling them with water until saturation. In a few days, suffering manifested in the plants, consisting of cigarette-like leaf rolling and leaf epinasty that culminated with complete loss of viability [43] (Figure 15). This not only demonstrates the high potential of waterlogging to cause stressful conditions for olive tree, but it shows once more that the cigarette-like leaf rolling is a highly unspecific sign of suffering in olive tree.

## 7. Common Threads and Specificities between OQDS and BTD Symptomatologies and Detection of the Agents

On the basis of what has been reported, distinguishing OQDS and BTD through rapid symptom observation remains difficult or impossible, especially in the latest phases of disease progression. The cigarette-like leaf rolling does not solve the problem, as it is not pathognomonic, and in fact, both OQDS and BTD are characterized by this symptom. Moreover, many other biotic/abiotic stresses—even mechanical damage such as pruning or branch/twig breakage—leading to an interruption of water conduction in the vessels wilt foliage according to this mode. Leaf tip desiccation can help to discriminate the two forms of decline, though this symptom is not pathognomonic. In fact: (i) it is consistently associated with early symptom progression in OQDS-affected trees and in XFP-artificially inoculated trees, and (ii) it has not been associated with pathogenic action of *Neofusicoccum* spp. after artificial infection [32,33]. However, just to underline the complexity of the matter, it is worth remembering that leaf scorch (i.e., leaf tip desiccation), in addition to twig dieback (cigarette-like leaf rolling and wilting), has been reported as a common symptomatic feature of twig and branch dieback of olive tree in California [42], a disease from whose etiology XF was excluded, and instead Botryosphaeriaceae were involved [36,42]. Leaf tip necrosis by *N. luteum* and *N. dimidiatum* can be easily distinguished from that of OQDS, as described above.

On the other hand, the reddish isles on leaf blades that can precede complete necrosis and desiccation in BTD-affected trees seem more typical of this type of decline. However, the reddish/brown/blackish bark canker and the wood discoloration is an especially important discriminating factor, as it characterizes BTD and *Neofusicoccum*-inoculated trees and is not incited by XFP. However, in this case, too, it has to be underlined that wood discoloration, even in the form of wedge-shaped cankers, is not strictly pathognomonic for Botryosphaeriaceae, as other fungal taxa are also able to incite such symptoms, for example, Dyatripaceae [85].

Table 1, on the next page, summarizes the symptomatic features of OQDS and BTD and other olive diseases, highlighting differences and similarities.

Obviously, in this scenario, the application of laboratory-based diagnostic methods targeted toward XFP and fungi is important in order to understand which type of syndrome is present. Several serological and molecular tests have been developed for XF detection, and specific differential recommendations have been given in the recently published “PM 7/24 (5) *Xylella fastidiosa*” [15] for diagnostic procedures to be applied in different conditions and for different aims: symptomatic and asymptomatic trees, testing from a known outbreak area or a buffer zone around an outbreak, testing for sub-species and sequence-type assignment. European regulation [86] established the molecular tests to be used for official XF detection: real-time PCR by Harper et al. [15,87] and Ouyang et al. [15,88], LAMP PCR [15] and conventional PCR by Minsavage et al. [15,89], which have been all extensively validated. Multi-locus sequence typing (MLST) by Yuan et al. [15,90] and conventional PCR by Hernandez-Martinez et al. [15,91] and Pooler and Hartung [15,92], each fully validated, have been established for XF sub-species assignment. Isolation on culture media is not recommended for XF as a diagnostic test due to its very low diagnostic sensitivity, namely, the difficulty in successfully isolating XF from the infected plant tissues [15].

Currently, only isolation on culture media is available as a method to ascertain which potentially pathogenic fungal species, apart from Botryosphaeriaceae, are associated with wood discoloration of declining olive trees. However, fungal isolation can also be widely prone to false negative detections; thus, it would be crucial to apply a metagenomic approach in order to integrate the results of the traditional culture-dependent fungal isolation.

Nevertheless, an effective molecular diagnosis is not yet fully conclusive: as stated above, we are determining that both XFP and wood-discoloration-causing fungi can co-habit the same trees. Thus, revealing whether XFP and fungi also act as a polyspecies in causing the olive decline is a crucial research focus. Key for this investigation should be to adapt the Koch postulate from the traditional one-pathogen/one-disease paradigm to a more flexible and multi-faceted experimental model that is able to demonstrate the pathogenic potential of the interactions between different microorganisms and their temporal succession, even in relation to the host plant reactivity.

## 8. Final Remarks

The nature of the olive decline in Salento still requires investigations in order to be fully understood. This takes for granted that XFP has an undeniable etiological involvement.

After having ascertained the strong similarity between the symptomatologies of OQDS and BTD, the question naturally arises as to whether we are facing a complex disease rather than multiple overlaying diseases.

Therefore, our conclusions cannot be conclusive, and instead, they raise a number of questions that can be useful to orient future research. Key questions are:Is XFP virulence tuned/assisted by other factors in relation to the olive decline occurring in Salento?Has the role of additional biotic and abiotic factors been underestimated in relation to the olive decline occurring in Salento?Which scenarios of interaction are currently underway among XFP and Botryosphaeriaceae and, possibly, what are the additional predisposing/inciting/contributing factors?

We feel that these questions should be taken into due consideration, especially in a research context in which the relationship between XF and the host microbiota is considered of great significance to unravel the so-called “conditional” lifestyle of XF, endophitism vs. pathogenicity [15,93]. Indeed, XF can infect more than 600 plant species, many of which are colonized endophytically. Meanwhile, XF expresses pathogenicity in only a limited number of species [2]. We have described above in detail how the demonstration of XFP pathogenicity in olive tree has been troublesome, as it has required very long times to be verified or did not yield substantial results [44,55]. At the same time, the bacterium has been found in stably asymptomatic olive trees. This strongly suggests that an endophytism can be adopted by XFP, and a switch in the micro-biosphere represented by the site–climate–microbiota–host plant would be necessary for the pathogenicity of XFP to be elicited and expressed.

Indeed, XF and Botryosphaeriaceae are well known both as aggressive plant pathogens and endophytic colonizers of a wide range of plant species, from agricultural crops to ornamental and forests hosts. They are thus not limited by any boundary. As endophytes, they can adopt a pathogenic lifestyle or increase their virulence when environmental stress conditions impact the host plants [94,95,96].

A holistic approach to shed more light on the nature of the olive decline occurring in Salento should be based on Koch postulate-based investigations readapted from the simple “one-pathogen” experimental model to “poly-species” and even “abiotic stressor-conditioned” experimental models. In Table 2, we summarize the possible scenarios of interaction of Botryosphaeriaceae with XFP in exacerbating symptom expression.

## Figures and Tables

**Figure 1 plants-12-03593-f001:**
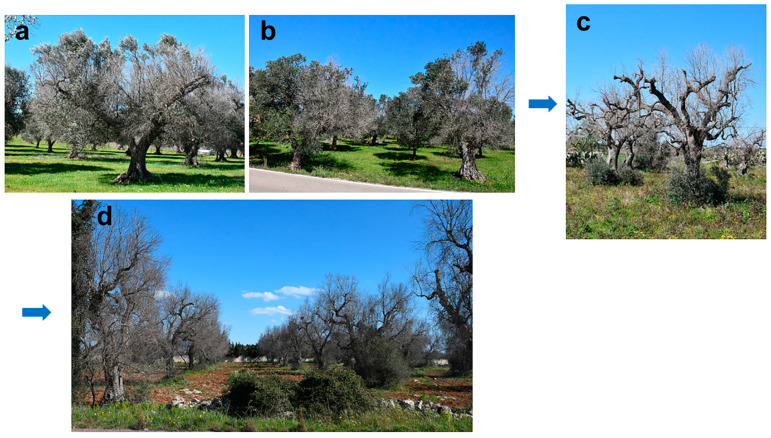
Advanced stages and final outcome of olive declines in Salento (Apulia). (**a**,**b**) Dieback severely affects large sectors of the canopy; (**c**) resprouting at the collar can occur in completely desiccated trees; (**d**) death of the whole tree is the final outcome.

**Figure 2 plants-12-03593-f002:**
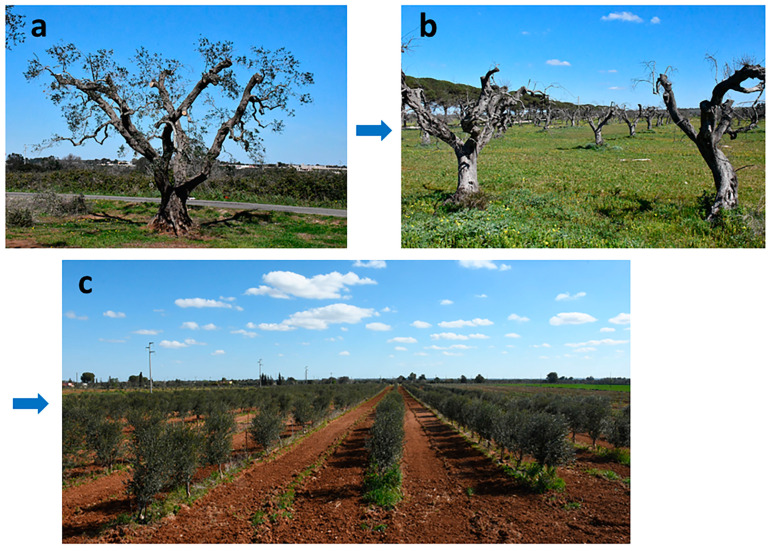
Attempts to remedy olive declines in Salento. (**a**) Growers resort to severe pruning to eliminate the dead parts of the tree scaffold, aimed at halting the disease progression; (**b**) however, the measure is unsuccessful, as dieback continues, and death inevitably occurs. (**c**) Plantings destroyed by decline are sometimes replaced with super-high-density groves of the cv FS-17, known to have a degree of resistance to XFP.

**Figure 3 plants-12-03593-f003:**
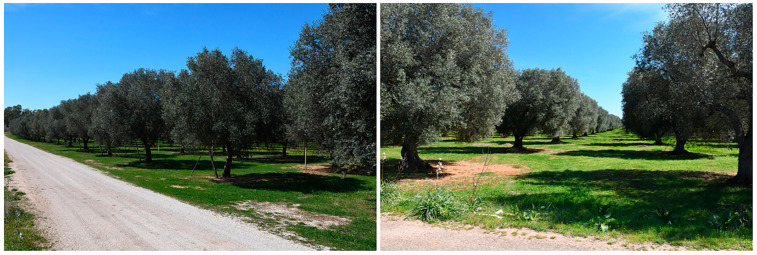
Groves of mature olive trees in Salento showing an optimal vegetative state, not affected by any decline symptoms, though located nearby to declining plantings.

**Figure 4 plants-12-03593-f004:**
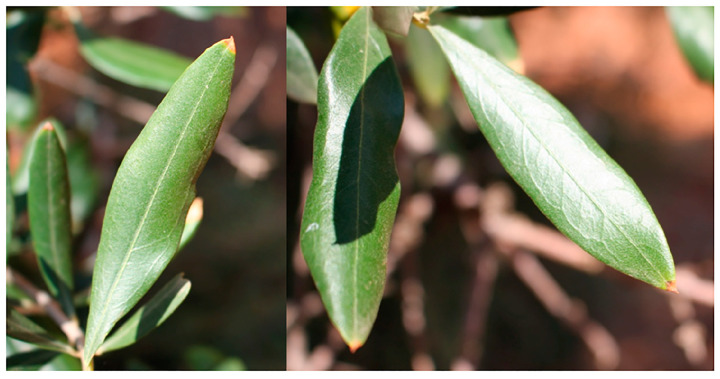
Leaf tip desiccation observed in spring on olive trees in Salento, upon the early-stage infection of *Xylella fastidiosa* subsp. *pauca* (XFP).

**Figure 5 plants-12-03593-f005:**
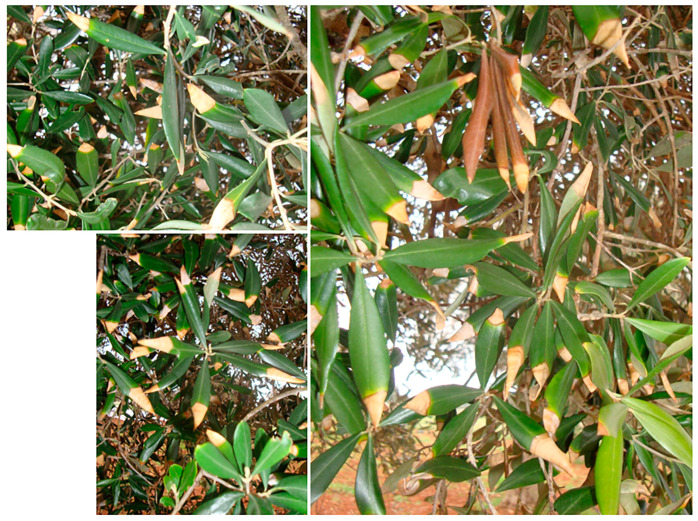
Progression of leaf tip desiccation caused by *Xylella fastidiosa* subsp. *pauca* (XFP) in advanced spring (pictures on the **left**) and leaf tip desiccation associated with collapse of the whole blade during summer (picture on the **right**) (Salento).

**Figure 6 plants-12-03593-f006:**
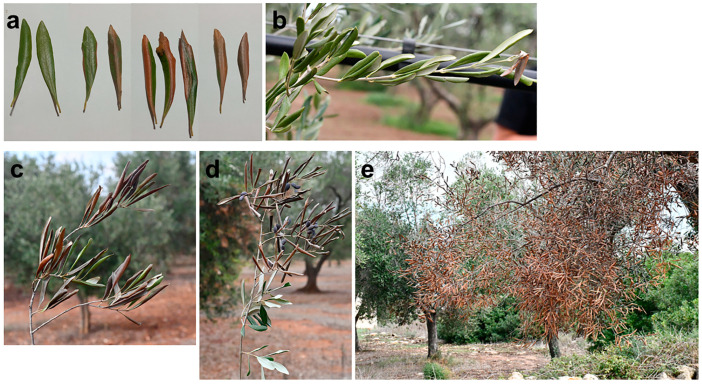
Symptom progression on leaves of olive trees in Salento infected with *Neofusicoccum* spp. and affected by branch and twig dieback (BTD). (**a**) Chlorotic and red-bronze to brown leaf blades; (**b**–**d**) downward leaf rolling; (**e**) overall view of a desiccated branch with cigarette-like wilted leaves.

**Figure 7 plants-12-03593-f007:**
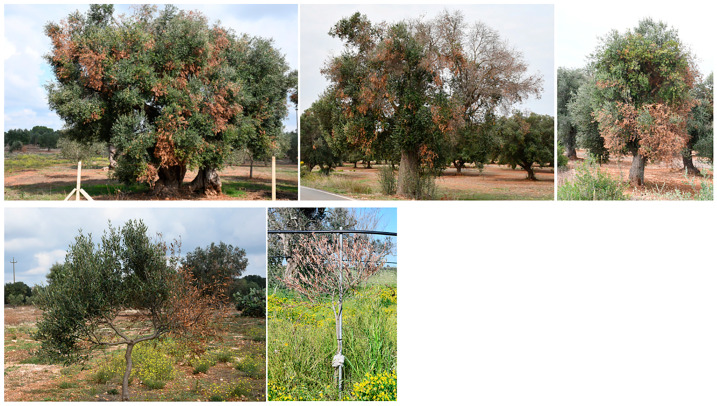
Mature and young olive trees in Salento infected with *Neofusicoccum* spp. and affected by severe branch and twig dieback (BTD).

**Figure 8 plants-12-03593-f008:**
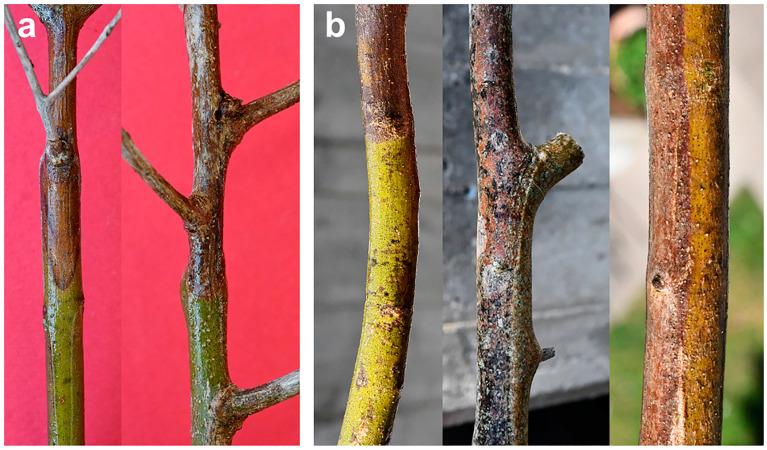
Bark canker with a basipetal progression at the base of the wilted twigs (**a**) and branches (**b**) in olive trees infected with *Neofusicoccum* spp. and affected by branch and twig dieback (BTD) in Salento.

**Figure 9 plants-12-03593-f009:**
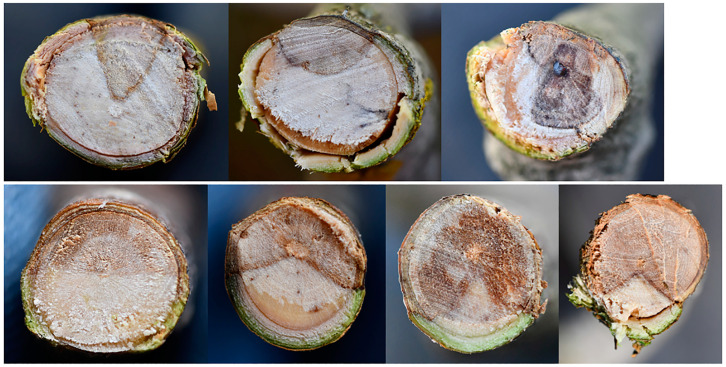
Wood discoloration in branches bearing wilted foliage in olive trees infected with *Neofusicoccum* spp. and affected by branch and twig dieback (BTD) in Salento (seen in cross-section). Note the wedge-shaped pattern of the discoloration.

**Figure 10 plants-12-03593-f010:**
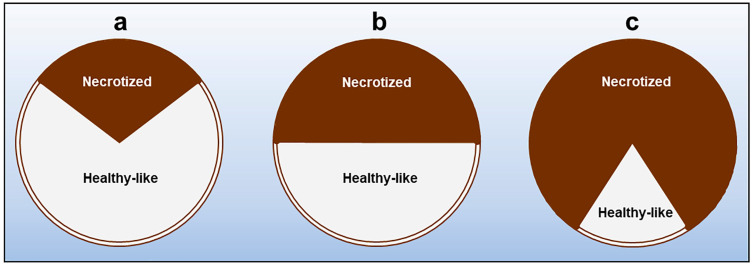
Graphic representation of different degrees of severity of wedge-shaped cankers in branches of olive trees affected by branch and twig dieback (BTD) (seen in cross section). Based on pathogenicity trials, *Neofusicoccum stellenboschiana* is able to cause necrotic patterns depicted in (**a**,**b**). On the other hand, *N. mediterraneum* has a higher virulence, being able to (almost) entirely girdle the stem circumference and to necrotize most of the internal xylem tissue (**c**). Invasion of xylem proceeds from bark via medullary rays. Thus, the rate of xylem colonization depends on the rate of bark girdling by the fungus.

**Figure 11 plants-12-03593-f011:**
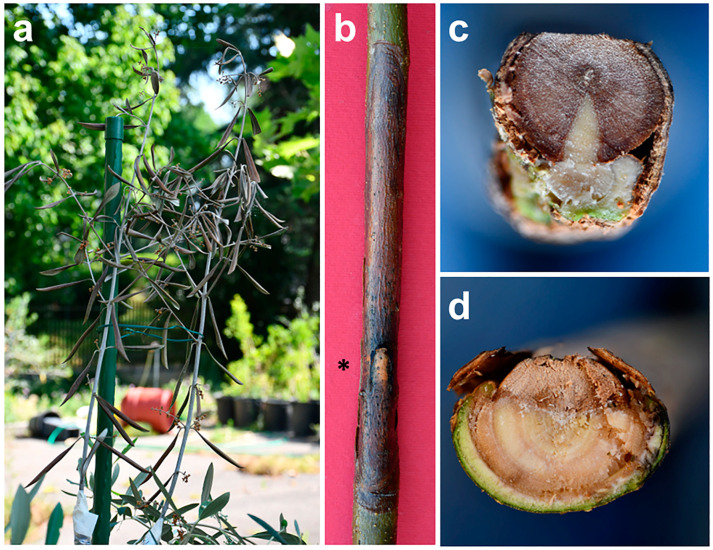
Symptoms occurring in olive trees inoculated with *Neofusicoccum mediterraneum* and *N. stellenboschiana* isolated from olive trees in Salento affected by branch and twig dieback (BTD): (**a**) wilting of twigs; (**b**) severe bark cankers in the stem of three-year-old trees (the black asterisk indicates the inoculation point); (**c**,**d**) wedge-shaped discoloration in the xylem caused by *N. mediterraneum* and *N. stellenboschiana,* respectively (seen in cross-section).

**Figure 12 plants-12-03593-f012:**
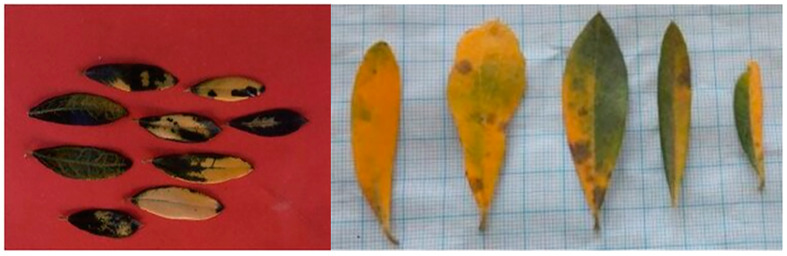
Foliar symptoms in olive tree infected with olive vein yellowing-associated virus (OVYaV). Image from Faggioli and Barba [75].

**Figure 13 plants-12-03593-f013:**
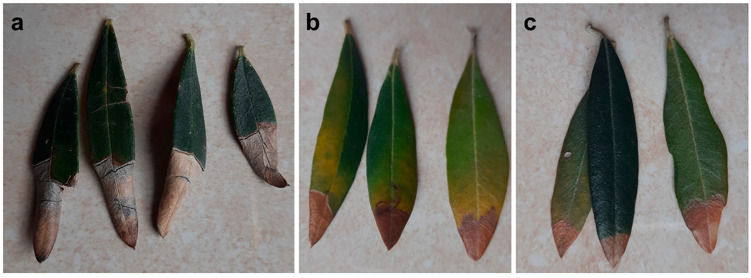
Symptoms of leaf tip desiccation (namely, apical leaf scorch) due to abiotic stress factors. (**a**) Symptoms caused by hot wind: the tip appears completely dry, shriveled and of ash color. (**b**) Symptoms caused by boron deficiency: note the extended chlorotic haloes bordering the desiccated areas of leaf tips. (**c**) Symptoms caused by potassium deficiency: note the absence of the chlorotic halo characterizing boron deficiency (in **b**). Courtesy: S. Pachioli, Istituto Tecnico Agrario “Cosimo Ridolfi”, Scerni, Italy.

**Figure 14 plants-12-03593-f014:**
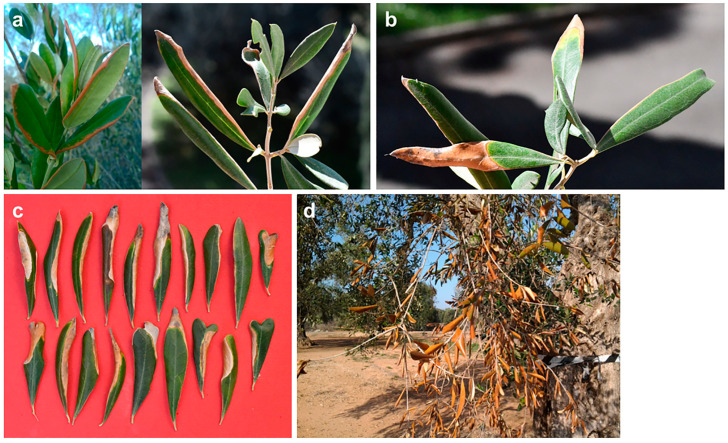
Olive leaves damaged by winter frost. (**a**) Initial marginal leaf scorch; (**b**,**c**) the scorch affects large areas of the leaf blades; (**d**) complete withering and leather color of the leaf blades some days after winter frost.

**Figure 15 plants-12-03593-f015:**
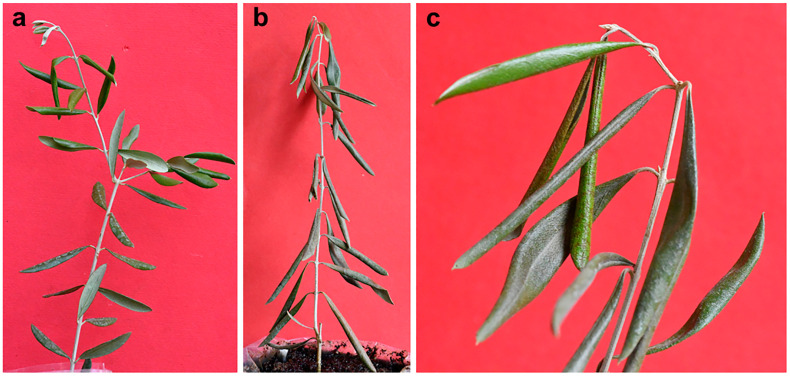
Symptoms of waterlogging obtained in one-year-old cuttings of olive tree cv Frantoio. (**a**) First symptoms of leaf rolling after six days of water saturation of the potted substrate; (**b**) plant close to death showing evident cigarette-like leaf rolling and leaf epinasty after 13 days’ water saturation; (**c**) close-up view of the cigarette-like leaf rolling symptom.

**Table 1 plants-12-03593-t001:** Diseases of olive tree sharing some important symptomatic features—leaf tip desiccation (i.e., apical leaf scorch), leaf chlorosis and yellowing, cigarette-like leaf rolling and wilting, dieback. The table serves for a preliminary symptom-based diagnosis, orienting the subsequent necessary laboratory-based diagnosis. Colored boxes identify the symptomatic features that distinguish olive quick decline syndrome (OQDS, by *Xylella fastidiosa* subsp. *pauca*) (in red) from branch and twig dieback (BTD, by *Neofusicoccum* spp. and possibly other fungi) (in green).

Symptoms Diseases and Their Agents	Leaf Yellowing	Leaf Mottling	Leaf Chlorosis	Leaf Tip Necrosis (Apical Leaf) Scorch)	Marginal Leaf Necrosis	Red-Bronze-Coalescing Patches in the Leaf	Cigarette-Like Leaf Rolling and Wilting	Leaf Fall	Bark Canker and Wood Discoloration	Dieback
OQDS *Xylella fastidiosa* subsp. *pauca*			✓	✓			✓	✓		✓
BTD Salento *Neofusicoccum* spp.			✓			✓	✓	✓	✓	✓
Parasitic brusca *Marthamyces panizzei*				✓ ^1^				✓		
Leaf necrosis and fruit rot *Neofusicoccum luteum*				✓ ^2^						
Canker and leaf scorch *Neoscytalidium dimidiatum*				✓ ^3^			✓	✓	✓	✓
Culture filtrates by: *Celerioriella prunicola*, *Phaeoacremonium* spp., *Neofusicoccum parvum*			✓				✓			
OVY OVYaV	✓									
OLY OLYaV	✓									
OYMD OYMDaV	✓	✓		✓				✓		✓
Hot and salty winds				✓ ^4^						
Boron deficiency ^5^			✓	✓						✓
Potassium deficiency ^6^				✓						
Winter frost					✓					
Waterlogging							✓			

^1^ Apothecia and picnidia develop in the necrotized tissue on the adaxial and abaxial leaf sides, respectively. ^2^ Picnidia develop in the necrotized tissue on the adaxial leaf side. ^3^ Picnidia develop in the necrotized tissue on the abaxial leaf side of the turkish cvs Gemlik and Ayvalık. ^4^ The necrotized leaf tissue appears completely dry, greyish and shriveled. ^5^ Additional symptoms associated with boron deficiency are: increased leaf thickness, leaf distortion, twig dieback accompanied by the occurrence of secondary shoots that lead to a “rosetta” formation and twig bark protuberance accompanied by phloem necrosis. ^6^ Additional symptoms associated with potassium deficiency are: small-sized leaves, leaf tips curled downward and twigs with short internodes.

**Table 2 plants-12-03593-t002:** *Xylella fastidiosa* subsp. *pauca* (XFP) and Botryosphaeriaceae are both able to cause severe damage to olive trees, namely, they can fulfill a pathogenic action independently of each other. Here, we want to outline some possible scenarios of interaction between the two microbial classes that have not been investigated so far but whose likelihood cannot be negated given the geographical overlapping of OQDS and BTD and the fact that mixed infections frequently occur. In all three cases, we have preferred to consider that XFP first colonize the host, and Botryosphaeriaceae thereafter. This does not mean that we exclude the reverse possibility.

Interaction Type 1	Interaction Type 2	Interaction Type 3
Asymptomatic XFP infections predispose to symptomatic infections by virulent Botryosphaeriaceae, which incite BTD. Botryosphaeriaceae might also carry out an additional action beyond that described above, namely, they break the equilibrium between XFP and the host plant by tuning “*the commensalism-parasitism setting knob*” toward parasitism, thus ultimately causing the commensal behavior of XFP “*to go badly”*.	XFP infections incite leaf tip desiccation or one–three-year-old twig/branch wilting and open the way for virulent Botryosphaeriaceae, which act as severe contributing factors, namely, they further aggravate the woody scaffold desiccation in a dieback mode. Low-virulent Botryosphaeriaceae and possibly other fungi can also occur and participate in the process, thus acting as a polyspecies. Similar to that described in interaction type 1, fungi might also carry out an additional action, namely, they in some way cause the XFP behavior to further exacerbate its virulence.	XFP infections primarily cause the whole canopy collapse in a dieback mode, also affecting the viability of the main trunk. This kicks off infections by virulent and low-virulent Botryosphaeriaceae (and possibly other fungi), which act as weak contributing factors, namely, they intervene in a disease state that has just dramatically compromised plant health. (The commensal behavior of XFP “*has gone badly”* independently of additional microbial players).

## Data Availability

Not applicable.

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
