# Peer review of "Xylella fastidiosa subsp. pauca, Neofusicoccum spp. and the Decline of Olive Trees in Salento (Apulia, Italy): Comparison of Symptoms, Possible Interactions, Certainties and Doubts"

_plants, 2023, doi:10.3390/plants12203593_

Round 1
Reviewer 1 Report
General comments:
Manuscript plants-2533979 describes the symptoms of different olive tree declines that occur in Salento (Apulia, Italy) and mention their confirmed or suspected causal agents. For the most part, the English writing is of good quality and the quality of images is excellent (except a very few). The choice of pictures to illustrate some of the symptoms caused by the causal agents is also remarkable.
However, beginning with the title, the content of the manuscript seems to lack a specific focus or purpose. The symptomatology associated with each agent (biotic or abiotic) is well described, but clear and direct comparisons among symptoms of different diseases is not provided (except in a very few sentences). Perhaps, the manuscript could have benefitted if a table comparing among the symptoms of the different diseases would have been included, giving emphasis to those that are characteristic of each particular disease, if any. My final conclusion after reading the manuscript is that it is very difficult to distinguish among them, which is recognized by the authors in some parts of the manuscript.
Given the difficulty of distinguishing among the different diseases, more emphasis should have been given to diagnostic methods. As a general rule, control methods must be established based on confirmative diagnosis (on pathogen identification or specific detection). In the end, I felt the authors were trying to transmit the idea that visual diagnosis could be the basis for the establishment of effective control methods. As a matter of fact, a brief description of the pathogens (in the cases of declines caused by biotic stresses) should be provided.
What was the chronology of the appearance of the different olive tree declines in Salento (Apulia)? Were those of fungal aetiology present in the area even before the detection of Xylella fastidiosa? Or, did they begin to appear after such a detection? What about the possibility that the presence of Xylella fastidiosa in the tree predispose it to fungal infection? Do those fungi cause disease in other tree species?
In some parts of the text, the authors (I believe unintentionally) raise doubts about the pathogenicity of Xylella fastidiosa on olive trees, based on the fact that infected trees do not exhibit disease symptoms. Perhaps, if they better discuss the biology of infection and how symptoms develop, with respect to the influence of bacterial population sizes in the infected plant, bacterial strain and plant genotypes as well as environment, could provide a clearer information (which was succinctly attempted in lines 215-218). Also, the concept of latent infection and the heterogeneous distribution of the pathogen in the plant tissue must be discussed, at least in the context of Xylella fastidiosa.
In this regard, given that different plant genotypes may respond differently to Xylella fastidiosa, comparison of results obtained in different geographic areas (countries) must be better approached by considering the plant and bacterial strain genotypes used in those studies or surveys. Also, environmental conditions must be provided in order to disclose possible differences that could explain the results.
Specific comments:
_ The tile could be improved to provide a more specific focus.
_ A few scientific names must be revised and, if necessary, replaced with those currently accepted; for instance, Pleurostomophora (Pleurostoma, according to MycoBank) and Stictis panizzei (Marthamyces panizzei, according to Mycobank).
_ The use of the words (concepts) virulence and pathogenicity must be revised; whereas virulence can be quantitative, pathogenicity is qualitative; the microorganism is pathogenic or not (there is nothing in between).
_ The use of “sensu stricto” in association with some terms need to be introduced for the readers understand exactly what the authors mean.
_ Line 73: revise the completeness of the sentence.
_ Line 86: “diversity and composition structure” is not clear. Which diversity? Which structure?
_ Line 91: although Pantoea agglomerans is frequently found as an endophyte in several plant species, it is also a bona fide plant pathogen causing disease in several host plants.
_ Lines 111-112: Why was this association not found in the literature? Could this information be built by searching the literature? It should be mentioned here.
_ Line 121, Could some numbers be provided here?
_ Line 319: What are the units for those numbers?
_ Line 333: Was this under natural conditions or upon artificial inoculation?
_ Line 380: “resulted susceptible to this symptomatic feature” does not make sense. The host is susceptible to the pathogen or the disease, but not to the symptoms.
_ Lines 527-529: Perhaps, with respect to the correct diagnosis of the diseases, this is the most important statement in the manuscript. But, it needs to be expanded to provide something useful for the readers or diagnosticians.
_ Lines 556-557: it would be very helpful for the readers, if the authors mention here that latent infections by Xylella fastidiosa are very commonly found in nature (similar to Pantoea agglomerans, which is commonly found as an endophyte, but it is also pathogenic).
The English language is of good quality, although a very few sentences and terms must be revised.
Author Response
We thank the referee 1 for the useful comments which helped us to improve the quality of the manuscript. Here below we report a point by point reply and refer to the manuscript lines where integration/modifications were made.
Massimo Pilotti
General comments:
Manuscript plants-2533979 describes the symptoms of different olive tree declines that occur in Salento (Apulia, Italy) and mention their confirmed or suspected causal agents. For the most part, the English writing is of good quality and the quality of images is excellent (except a very few). The choice of pictures to illustrate some of the symptoms caused by the causal agents is also remarkable.
However, beginning with the title, the content of the manuscript seems to lack a specific focus or purpose. The symptomatology associated with each agent (biotic or abiotic) is well described, but clear and direct comparisons among symptoms of different diseases is not provided (except in a very few sentences). Perhaps, the manuscript could have benefitted if a table comparing among the symptoms of the different diseases would have been included, giving emphasis to those that are characteristic of each particular disease, if any. My final conclusion after reading the manuscript is that it is very difficult to distinguish among them, which is recognized by the authors in some parts of the manuscript.
- Author response: Yes, I fully agree. I included such table (see page 21 of the manuscript).
Given the difficulty of distinguishing among the different diseases, more emphasis should have been given to diagnostic methods. As a general rule, control methods must be established based on confirmative diagnosis (on pathogen identification or specific detection). In the end, I felt the authors were trying to transmit the idea that visual diagnosis could be the basis for the establishment of effective control methods. As a matter of fact, a brief description of the pathogens (in the cases of declines caused by biotic stresses) should be provided.
- Author response: Yes, you’re fully right. I added a part to outline diagnosis targeted to XF and detection possibilities and perspectives for fungi. See lines 722 to 750
What was the chronology of the appearance of the different olive tree declines in Salento (Apulia)? Were those of fungal aetiology present in the area even before the detection of Xylella fastidiosa? Or, did they begin to appear after such a detection?
- Author response: OQDS has been reported in Salento in 2013 but appeared likely between 2008 and 2010, in a restricted area of the Ionian coast of the province of Lecce (see citation 8 by Martelli et al. 2016, in the manuscript). Relating to BTD, Neofusicoccum mediterraneum was reported in Apulia (Lecce province) in 2008 in olive trees, but only in the drupes and with a low occurrence (less than 1% of the examined drupes) (see Lazzizera et al. Morphology, phylogeny and pathogenicity of Botryosphaeria and Neofusicoccum species associated with drupe rot of olives in southern Italy. Plant Path. 2008, 57, 948–956). Also other botryospaeriaceae were reported in this article.Thus, we consider that the two pathogen classes arose contemporaneously in the Salento territory. Anyway, this also emerges in the introduction of the proof as the examination of the first OQDS samples yielded several fungal species potentially pathogenic to olive tree, which however were judged as not relevant for decline comprehension.
See also lines 47-51 of the manuscript which report that fungi, Botryosphaeriaceae included were detected since the appearance of OQDS (citation 8, 12, 19).
What about the possibility that the presence of Xylella fastidiosa in the tree predispose it to fungal infection?
- Author response: Yes, we are really aware of a possibility of interaction between XFP and Botryosphaeriaceae and possibly other fungi. We stress this point throughout the introduction, in the new chapter “2 Brief outlines of pathogenicity of Xylella fastidiosa subsp. fastidiosa in the Pierce’s disease of grapevine: can it be a teaching?” page 9, and in the “conclusive remarks” page 21 and 22 where we also present the (new) table 2 where we highlight the predisposition scenario of XF-infected trees in relation to subsequent infection by Botryosphaeriaceae.
Do those fungi cause disease in other tree species?
- Author response: this is briefly explained in lines 780-784
In some parts of the text, the authors (I believe unintentionally) raise doubts about the pathogenicity of Xylella fastidiosa on olive trees, based on the fact that infected trees do not exhibit disease symptoms. Perhaps, if they better discuss the biology of infection and how symptoms develop, with respect to the influence of bacterial population sizes in the infected plant, bacterial strain and plant genotypes as well as environment, could provide a clearer information (which was succinctly attempted in lines 215-218). Also, the concept of latent infection and the heterogeneous distribution of the pathogen in the plant tissue must be discussed, at least in the context of Xylella fastidiosa.
In this regard, given that different plant genotypes may respond differently to Xylella fastidiosa, comparison of results obtained in different geographic areas (countries) must be better approached by considering the plant and bacterial strain genotypes used in those studies or surveys. Also, environmental conditions must be provided in order to disclose possible differences that could explain the results.
- Author response: I think that I carefully considered all these points and integrated the text accordingly. See lines 273-284, 296-324, the whole (new) chapter lines 328-406.
Specific comments:
- Perhaps, the manuscript could have benefitted if a table comparing among the symptoms of the different diseases would have been included, giving emphasis to those that are characteristic of each particular disease, if any.
Author response: Yes, I fully agree. I included such table (see page 20 of the manuscript).
- The tile could be improved to provide a more specific focus.
- A few scientific names must be revised and, if necessary, replaced with those currently accepted; for instance, Pleurostomophora(Pleurostoma, according to MycoBank) and Stictis panizzei (Marthamyces panizzei, according to Mycobank).
Author response: Ok we corrected (lines 57, 513)
- The use of the words (concepts) virulence and pathogenicity must be revised; whereas virulence can be quantitative, pathogenicity is qualitative; the microorganism is pathogenic or not (there is nothing in between).
- The use of “sensu stricto” in association with some terms need to be introduced for the readers understand exactly what the authors mean.
Author response: we preferred to remove “sensu strictu” from line 95 and substitute with “primary”. In line 154 “sensu stricto” is referred to the term “decline” as Synclair or Manion defined. We explain in line 154-155.
- Line 73: revise the completeness of the sentence.
Author response: OK, see line 84-86.
- Line 86: “diversity and composition structure” is not clear. Which diversity? Which structure?
Author response: sorry I don’t understand..they are referred to the leaf microbiota…
- Line 91: although Pantoea agglomerans is frequently found as an endophyte in several plant species, it is also a bona fideplant pathogen causing disease in several host plants.
Author response: OK, now I’ve specified “…non pathogenic species for olive tree..” (line 103)
- Lines 111-112: Why was this association not found in the literature? Could this information be built by searching the literature? It should be mentioned here.
Author response: Sorry, I’ve found an important paper by Krugner et al. 2014 which surely deserve to be cited. Please see how I modified and what I added in lines 126-138.
- Line 121, Could some numbers be provided here?
- Line 319: What are the units for those numbers?
Author response: no units, it is a ratio (lines 483-485 in the revised proof)
- Line 333: Was this under natural conditions or upon artificial inoculation?
Author response: artificial inoculation, I have specified in the new text (line 489)
- Line 380: “resulted susceptible to this symptomatic feature” does not make sense. The host is susceptible to the pathogen or the disease, but not to the symptoms.
Author response: I eliminated “resulted susceptible to this symptomatic feature” and substituted with “developed this symptom” (line 544)
- Lines 527-529: Perhaps, with respect to the correct diagnosis of the diseases, this is the most important statement in the manuscript. But, it needs to be expanded to provide something useful for the readers or diagnosticians.
Author response: Yes I do agree. We expanded this part, see lines 722-742.
- Lines 556-557: it would be very helpful for the readers, if the authors mention here that latent infections by Xylella fastidiosaare very commonly found in nature (similar to Pantoea agglomerans, which is commonly found as an endophyte, but it is also pathogenic).
Author response: We express this concept in the conclusions: lines 765-779.
Reviewer 2 Report
This paper has reviewed over 70 relevant literatures, which have studied the contents shown on the title. Although I do not know much on olive, as a plant pathologist, we can not judge the possible interactions via such simple comparison of symptoms summarised by the authors. Indeed it's worthwhile to do such a review, in order to make readers clearly understand the points, it's suggested to reorganize it. To my understanding, those symptoms caused by biotic factors have been experimentally confirmed for most, of which bacteria-like organism Xylella fastidiosa is xylem-limited, unlike fungi. Therefore firstly, the distribution parts of the causing pathogens should be reviewed, is there any insect or mite vector? secondly the time frame should considered for each factor with potential affect, since some are the major ones, some might be with weak parasitism. Anyway, the field symptoms are affected by a lot of factors, which are comprehensive phenotypes and complicated. It's hard to make conclusion by such comparisons.
Author Response
Comments and Suggestions for Authors
This paper has reviewed over 70 relevant literatures, which have studied the contents shown on the title. Although I do not know much on olive, as a plant pathologist, we can not judge the possible interactions via such simple comparison of symptoms summarised by the authors. Indeed it's worthwhile to do such a review, in order to make readers clearly understand the points, it's suggested to reorganize it. To my understanding, those symptoms caused by biotic factors have been experimentally confirmed for most, of which bacteria-like organism Xylella fastidiosa is xylem-limited, unlike fungi. Therefore firstly, the distribution parts of the causing pathogens should be reviewed, is there any insect or mite vector? secondly the time frame should considered for each factor with potential affect, since some are the major ones, some might be with weak parasitism. Anyway, the field symptoms are affected by a lot of factors, which are comprehensive phenotypes and complicated. It's hard to make conclusion by such comparisons.
Author_response: The revised version includes a new Table (Table 1) that summarizes all the main symptoms incited by the biotic factors as well as the abiotic ones. From this Table, it clearly appears which are the differences among the various agents discussed in the review. It is well known that Xylella fastidiosa is an insect-trasmitted bacterium but the features concerning the vector are not within the aim of this review. For sure, Xylella fastidiosa and the Botryospheraceae are the main cause of damage in Apulia and they are discussed to a larger extent compared to the other ones, including in the Abstract. We agree that the final symptom is affected by several aspects that are not easy to point out. However, for some of them we are currently investigating.
Massimo Pilotti
Reviewer 3 Report
In my opinion the article is well written, concerns important problem for olives and may be interesting for researchers in the given field. Especially I appreciate the colour photos of symptoms caused by different factors. Thus the article can have educational value as well.
I found only several minor mistakes:
I haven´t found any order in Botryosphaeriaceae writting throughout the article. Sometimes it is in italics, sometimes not. Perhaps more complicated expression botryosphaeriaceous fungi (as on line 103) or b. species could be used if scientific family name is not need to be used.
cultivar cv probably cv. (multiple)
189: pauca in italics
410: "normal" citations (Savino et al., Caruso et al.) instead of numbers. Since references are not in alphabetical order it makes more difficult to find these two among them.
Phytopathology in references: once it is abbreviated Phytopath (6), in other case it is as full name (29).
First letters in article titles in references should not be capitalized (6, 71).
Author Response
Comments and Suggestions for Authors
In my opinion the article is well written, concerns important problem for olives and may be interesting for researchers in the given field. Especially I appreciate the colour photos of symptoms caused by different factors. Thus the article can have educational value as well.
- Author response: Thank you very much for appreciating our article!
I found only several minor mistakes:
I haven´t found any order in Botryosphaeriaceae writting throughout the article. Sometimes it is in italics, sometimes not. Perhaps more complicated expression botryosphaeriaceous fungi (as on line 103) or b. species could be used if scientific family name is not need to be used.
- Author response: I corrected and used only “Botryosphaeriaceae” troughout the text
cultivar cv probably cv. (multiple)
- Author response: I corrected and used only “cv” (or “cvs”) before the name of the cultivar
189: pauca in italics
- Author response: I put in italics “pauca” throughout the text.
410: "normal" citations (Savino et al., Caruso et al.) instead of numbers. Since references are not in alphabetical order it makes more difficult to find these two among them.
Phytopathology in references: once it is abbreviated Phytopath (6), in other case it is as full name (29).
First letters in article titles in references should not be capitalized (6, 71).
- Author response: Yes thank you, we checked all the list of citations: “Phytopathology” must not abbreviated (according to the rules..). We also removed the capitalized letters from the titles of the articles.
Massimo Pilotti
Reviewer 4 Report
This is just a review and from that point of view it is a good summary of the published literature. Overall I would rate it average in all respects. The authors pointed out the problems but failed to provide new innovative ideas to solve the problem.Lines 29, 189 pauca should be in italics. Line 380 author citation should be inserted. Line 410 numerical citations should be used instead if authors & year
Author Response
Comments and Suggestions for Authors
This is just a review and from that point of view it is a good summary of the published literature. Overall I would rate it average in all respects. The authors pointed out the problems but failed to provide new innovative ideas to solve the problem.
- Author response: Thank you for appreciating our review. This is a quite complex matter and we underline that additional research is needed to elucidate completely the nature of decline in Salento. Specifically, the precise aim of this work is to point out the weaknesses of the knowledge acquired so far and what is still lacking in order to fill the gaps.
Lines 29, 189 pauca should be in italics.
- Author response: we have corrected “pauca” throughout the text
Line 380 author citation should be inserted.
- Author response: we corrected (now line 544)
Line 410 numerical citations should be used instead if authors & year
- Author response: we corrected (now line 574)
Massimo Pilotti
Reviewer 5 Report
This is a very good and comprehensive treatment to this very common and destructive disease to the olive plantations. Some years ago, same disease syndromes were also observed in Jordan, Middle East.
It is very important to carry-on with this work utilizing the new biotechnology tools and techniques ascertaining on a time-based follow-up on the early stages of microbial (bacteria and fungi), and environmental interactions behind this destructive disease.
Author Response
Comments and Suggestions for Authors
This is a very good and comprehensive treatment to this very common and destructive disease to the olive plantations. Some years ago, same disease syndromes were also observed in Jordan, Middle East.
It is very important to carry-on with this work utilizing the new biotechnology tools and techniques ascertaining on a time-based follow-up on the early stages of microbial (bacteria and fungi), and environmental interactions behind this destructive disease.
Author response: Thank you very much for appreciating our work! Happy to know more about syndrome in Jordan.
Massimo Pilotti
Reviewer 6 Report
The manuscript of Xylella fastidiosa subsp. pauca, Neofusicoccum spp. and the declines of olive trees in Salento (Apulia, Italy): comparison of symptoms, possible interactions, certainties and doubts by Scortichini et al., discussed Olive Quick Decline Syndrome (OQDS) and Branch Dieback Disease (BTD) caused by different fungus, biotic agents and abiotic agents in details. Those comprehensive explorations provide invaluable guidance for accurate symptomatic diagnoses, contributing significantly to the productivity and sustainability of olive tree agriculture. I agree to accept this review in present form.
Minor editing of English language required.
Author Response
The manuscript of Xylella fastidiosa subsp. pauca, Neofusicoccum spp. and the declines of olive trees in Salento (Apulia, Italy): comparison of symptoms, possible interactions, certainties and doubts by Scortichini et al., discussed Olive Quick Decline Syndrome (OQDS) and Branch Dieback Disease (BTD) caused by different fungus, biotic agents and abiotic agents in details. Those comprehensive explorations provide invaluable guidance for accurate symptomatic diagnoses, contributing significantly to the productivity and sustainability of olive tree agriculture. I agree to accept this review in present form.
- Author response: Thank you a lot for your appreciation!
Massimo Pilotti